# Fabrication of High Aspect Ratio Nano-Channels by Thermal Nano-Imprinting and Parylene Deposition

**DOI:** 10.3390/mi14071430

**Published:** 2023-07-16

**Authors:** Kun Yang, Zhifu Yin, Lei Sun

**Affiliations:** 1Shanxi Key Laboratory of Micro Nano Sensors & Artificial Intelligence Perception, College of Information and Computer, Taiyuan University of Technology, Taiyuan 030024, China; yangkun01@tyut.edu.cn; 2State Key Laboratory of Electrical Insulation and Power Equipment, Xi’an Jiaotong University, Xi’an 710049, China; yinzf@jlu.edu.cn; 3The State Key Laboratory of Refractories and Metallurgy, Wuhan University of Science and Technology, Wuhan 430081, China

**Keywords:** nano-channels, high aspect ratio, SU-8 photoresist, Parylene deposition

## Abstract

A low-cost method of fabrication of high aspect ratio nano-channels by thermal nano-imprinting and Parylene deposition is proposed. SU-8 photoresist nano-channels were first manufactured by thermal nano-imprinting, and Parylene deposition was carried out to reduce the width of the nano-channels and increase the aspect ratio. During the process, the side walls of the SU-8 nano-channels were covered with the Parylene film, reducing the width of the nano-channels, and the depth of the channels increased due to the thickness of the Parylene film deposited on the surface of the SU-8 nano-channels, more so than that at the bottom. The influence of Parylene mass on the size of nano-channels was studied by theoretical analysis and experiments, and the deposition pressure of Parylene was optimized. The final high aspect ratio nano-channels are 46 nm in width and 746 nm in depth, of which the aspect ratio is 16. This simple and efficient method paves the way for the production of high aspect ratio nano-channels.

## 1. Introduction

In recent years, with the development of nanotechnology, nanofluidic chip technology has emerged. The minimum dimension of the chip goes down from the microscale to the nanoscale, which changes the surfaces of the channels significantly. Accordingly, new flow phenomena occur, such as the overlay of the double electrode layer, the increase in viscosity, the increase in the resistance of the fluid in the nano-channels, and the decrease in the dielectric constant. Researchers are attracted by these new phenomena; nano-fluidic chips provide a new research method to conduct biological science and technology research, such as genomic sequencing, detection of macromolecules, and ion sample preconcentration.

The nano-channel, which has a strong impact on the property of the nano-fluidic chip, is the key structure of the nano-fluidic chip. It plays an irreplaceable role in many fields, such as DNA sequencing [1,2,3,4], biochemical analysis [5,6,7,8,9], and environment detection [10,11,12]. Therefore, the fabrication methods for nano-channels have been of wide concern in nano-manufacturing.

At present, the fabrication methods for nano-channels mainly include high-resolution machining technology [13,14,15], sacrificing layer technology [16,17], the crack method [18,19], PDMS deformation [20,21], molecular self-assembly [22,23,24], and nano-imprinting technology [25,26,27]. The high-resolution machining technology mentioned above uses nanoscale resolution to expose or directly write nano-channels on the substrate, which provides the advantages of flexibility, controllability, high accuracy, and good quality. This method can manufacture high-quality nano-channels below 10 nm in size and is capable of processing complex nano-structures. However, it relies on expensive equipment and high requirements for experimental conditions, which leads to high manufacturing costs and low production efficiency of nano-channels. In sacrificing layer technology, nano-channels were obtained by making nanosized sacrificial layer structures, then growing other materials on top of them, and finally removing the sacrificial layer. However, the sacrificial layer corroded for a long time, the stress was released during the removal of the sacrificial layer, and the mechanical deformation of the nano-channels occurred, resulting in an uneven cross-section size of the nano-channels. In addition, when the nano-channel was long, the sacrificial layer was not easily completely corroded, and the reactants remained inside the nano-channels, causing partial or even complete blockage of the nano-channels. For the methods of crack and PDMS deformation, with the high elasticity of PDMS, the phenomena of collapse and folding are prone to occur during processing, and some researchers have made use of these phenomena to create nano-channels. This method is cheaper to produce, but the shape and size of the nano-channels cannot be controlled accurately due to the ungovernability of the crack shape and the PDMS deformation. Molecular self-assembly is a process in which individual molecules spontaneously compose new molecular structures. Using molecular self-assembly technology, nano-channels with a size of several nanometers can be formed. The production of nano-channels by the molecular self-assembly method is low in cost, but it is also disorderly.

Nano-imprint technology is a kind of nano pattern reproduction method, which is divided into thermal nano-imprinting technology and ultraviolet-curing nano-imprinting technology. Thermal nano-imprinting technology is performed as follows: spin a layer of polymer on the substrate, soften and deform the polymer by heating and pressurizing, fill the mold pattern to form a nano-structure, and then remove the residual thin layer of polymer through etching technology to expose the substrate material. This method has the advantages of high resolution, low cost, and high efficiency, and is one of the most important methods to fabricate nano-channels. However, the damage caused by demolding during thermal nano-imprinting greatly affects the forming qualities of nano-channels, especially for high aspect ratio structures.

To sum up, the fabrication of nano-channels is hindered by its high cost and complicated and time-consuming processes, particularly for high aspect ratio nano-channels. As such, the development of novel fabrication methods with low cost and high efficiency is of great practical significance. In this paper, a low-cost fabrication method for high aspect ratio nano-channels by thermal nano-imprinting and Parylene deposition is proposed. The SU-8 photoresist nano-channels were first produced by thermal nano-imprinting, and the high aspect ratio nano-channels were fabricated by Parylene deposition. During the Parylene deposition process, the side walls of the SU-8 nano-channels were covered with Parylene film, reducing the width of the nano-channels, and due to the thickness of the Parylene film deposited on the surface of the SU-8 nano-channels, more than that at the bottom, the depth of the nano-channels increased. The influence of the Parylene mass on the film thickness and the effect of the deposition pressure on the film roughness were investigated. The final nano-channels thermal bonded with SU-8 photoresist are 46 nm in width and 746 nm in depth, of which the aspect ratio is 16. This method is simple, low-cost, does not rely on expensive processing equipment, and can achieve different sizes of nano-channel manufacturing below 100 nm, including high aspect ratio nano-channels, which has potential application value for the development of nano-fluidic chips.

## 2. Experiments

### 2.1. Fabrication Process of Nano-Channels

Figure 1 shows the fabrication process of nano-channels by thermal nano-imprinting and Parylene deposition. First, the SU-8 photoresist (MicroChem, SU-8 2015) was spin-coated on a silicon substrate (1.5 cm × 2 cm) at a speed of 4000 rpm for 30 s, followed by pre-baking at 85 °C for 30 min (Figure 1a,b). Then, the SU-8 photoresist was thermal imprinted by a silicon nano-mold at 0.1 Mpa by two clamps in an oven at 85 °C for 5 min (Figure 1c), and the silicon nano-mold was removed after cooling (Figure 1d). After that, the high aspect ratio nano-channels were produced by chemical vapor deposited (CVD) with Parylene N (Suzhou Yacoo Science Co., Ltd., Suzhou, China) on the SU-8 nano-channels. For the next step, SU-8 photoresist 2015 was spin-coated on the PDMS substrate at a speed of 4000 rpm for 30 s, followed by pre-baking at 85 °C for 30 min to produce a cover SU-8 layer (Figure 1f). Then, the nano-channels were thermal bonded with the cover SU-8 layer at 50 °C for 5 min. (Figure 1g). Before the thermal bonding, the SU-8 photoresist cover plate and Parylene nano-channels were treated with oxygen plasma (Emitech K1050X) under these parameters: 30 s at a chamber power of 30 W, to improve the surface energy, which increases the bonding strength and ensures complete bonding. Finally, the PDMS was removed from the cover SU-8 layer, and the high aspect ratio nano-channels were fabricated (Figure 1h).

### 2.2. Parylene Deposition

Parylene materials can be divided into Parylene N, Parylene C, Parylene D, Parylene HT, and Parylene F, according to different molecular structures [28]. Among them, Parylene N has good coverage during deposition, and can be deeply covered on the surfaces of various structures, including deep-hole structures and channels with high aspect ratios, etc. Furthermore, it has a low deposition temperature and low stress, and the deposited film has good uniformity and densification, and low surface roughness. Therefore, this paper’s authors chose to grow Parylene N to reduce the size of the nano-channels.

The Parylene film described in this paper was chemical vapor deposited (CVD) by a Parylene deposition system (PDS2010, Labcoter, Indianapolis, IN, USA), which mainly includes an evaporation chamber, pyrolysis chamber, deposition chamber, cooling system, and vacuum pump. Parylene original material was placed in the evaporation chamber, and the substrate was placed in the deposition chamber. The temperature of the evaporation chamber rose when the vacuum value of the deposition chamber reached the set value P0. At 150 °C, the Parylene original material vaporized into gas and entered the pyrolysis chamber, where it decomposed into monomers at the high temperature of 650 °C. Then, the Parylene monomers entered the deposition chamber, raising the pressure of the deposition chamber. When the pressure of the deposition chamber reached the set value, Parylene began to deposit. By the time the lack of Parylene original material in the evaporation chamber reduced the evaporation rate, leading to a drop in the chamber pressure, the Parylene film was completed deposited; the specific deposition parameters are shown in Table 1.

## 3. Results

### 3.1. Influence of Parylene Mass on Nano-Channel Size

When the Parylene film was deposited on the SU-8 photoresist nano-channels, the Parylene covered the side walls and bottoms of the channels, thereby reducing the size of the nano-channels. The size of the nano-channel is determined by the thickness of the Parylene film, which mainly depends on the quality of the Parylene raw material. During this experiment, 0.05 g, 0.08 g, 0.10 g, and 0.12 g of Parylene N material were used for CVD, and the thickness of the Parylene film was measured by an ellipsometer (M-2000DI, J.A.Woollam, Lincoln, NE, USA).

Figure 2 shows the relationship between the thickness of the Parylene film and the mass of the Parylene material, where the thickness of the Parylene film is the average value of experimental measurements repeated 3 times. It can be seen that the thickness of the Parylene film increased with the Parylene mass increasing. Theoretically, the thickness of the Parylene film and the Parylene mass are proportional to each other. In fact, when the Parylene material mass is less, the thickness of the Parylene film is thin, and when the Parylene raw material is sufficient, the thickness of the Parylene film is basically proportional to the quality of the raw material. This is due to some losses during the deposition process as well as residual material.

Figure 3 shows the morphology of the SU-8 photoresist nano-channels fabricated by thermal nano-imprinting shown in Figure 1d, which are 328 nm in width and 756 nm in depth. The sizes of the nano-channels were reduced by Parylene deposition. Ideally, if the thickness of the Parylene film deposited on the side walls and bottoms of the SU-8 nano-channels is equal to the thickness deposited on the SU-8 surfaces, then the relationship between the nano-channel size and the thickness of the Parylene film is as shown in Figure 4, which can be expressed by the following formula:*w* = *w*_0_ − 2*t*
(1)
*h* = *h*_0_(2)
where *w* and *h* are the final width and depth of the channel, respectively, *w*_0_ and *h*_0_ are the initial width and depth of the channel, respectively, and *t* is the thickness of the deposition of the Parylene film. Therefore, we can plot the theoretical relationship between channel size and Parylene mass.

In theory, if the thickness of the film deposited on the side wall and bottom is equal to the thickness deposited on the surface, the width of the nano-channel will become smaller and smaller, and the depth will remain the same. Therefore, the depth-to-width ratio of the nano-channel will increase with the increase of the Parylene mass. When the thickness of the Parylene film is two times greater than the width of the nano-channel, the nano-channel will be completely filled.

Parylene film growth was performed on the SU-8 nano-channels by CVD, and the morphology of the nano-channels was observed by scanning electron microscope (SEM, Olympus IX71, Tokyo, Japan), as shown in Figure 5. It can be seen that the Parylene film was well covered on the bottom and side walls of the SU-8 photoresist nano-channels. However, due to the high fracture elongation of Parylene film, the fracture elongation phenomenon shown in the figure occurred during the preparation of the cross-section electron microscope samples. Therefore, the thickness of the Parylene film deposited on the side wall and bottom cannot be observed in Figure 5, but the size of the nano-channels could be obtained.

The experimental relationship curve between the size of the nano-channels and the Parylene mass is shown in Figure 6. It can be seen that with the increase in the Parylene mass, the width of the nano-channel gradually decreases. This is because the side wall of the nano-channel is covered with the Parylene film, reducing the width of the nano-channel. The thicker the Parylene film, the smaller the width of the nano-channel. Compared with the experimental curve and the theoretical curve, both trends are the same.

When the Parylene mass was 0.05 g, the width of the nano-channels obtained by experiment was basically consistent with the theory. However, when the Parylene material mass increased, the width of the nano-channel became narrower; the difficulty of Parylene molecules entering the nano-channel was increased. Therefore, the probability of Parylene deposition on the side wall decreased, resulting in the actual reduction of the width to a lesser degree than the theoretical value. Therefore, when the Parylene mass was 0.08 g, the width of the nano-channels obtained by the experiment was wider than the theoretical value. When the Parylene was 0.1 g, the thickness of the Parylene film (*t*) was 206 nm (as shown in Figure 2), and 2*t* was larger than the SU-8 nano-channel width (*w*_0_ = 326 nm); the nano-channel should have been completely filled. However, in the actual experiment, the probability of the Parylene molecules entering the nano-channel and being deposited on the side wall of the channel was smaller than that of deposition on the surface of SU-8. Therefore, the thickness of the Parylene film on the side wall of the nano-channel was smaller than that in Figure 2, resulting in the nano-channel being incompletely filled, leaving the nano-channel as shown in Figure 5c.

Theoretically, if the thickness of the Parylene film deposited on the surface of an SU-8 nano-channel is the same as the thickness of the film deposited at the bottom of the SU-8 nano-channel, the depth of the nano-channel should be unchanged, but it in fact increased and then decreased with the addition of Parylene. This is because, when there is less Parylene material, the thickness of the Parylene film deposited on the surfaces of the SU-8 nano-channels was more than that at the bottoms, resulting in the distance from the bottoms of the nano-channels to the upper surfaces being larger, that is, the depth of the channels increased (Figure 5a,b). When there was more Parylene material, the Parylene growing at high temperature had a certain fluidity, and the Parylene attached to the side wall flowed to the bottom of the channel. As a result, more Parylene filled the bottom of the nano-channel, causing the channel depth to decrease (Figure 5c). When the Parylene material continued to increase, the sum of the thickness of the Parylene film deposited on the side wall of the channel was greater than the original width of the channel, resulting in the channel being filled with Parylene and blockage (Figure 5d).

Figure 7 shows the SEM images of multiple nano-channels after 0.08 g Parylene growth on the SU-8 photoresist nano-channels. It can be seen that the morphology of the nano-channels is intact, and the size of the nano-channels from left to right is: 95 nm (width), 802 nm (depth); 92 nm (width), 783 nm (depth); 88 nm (width), 798 nm (depth); 98 nm (width), 823 nm (depth); 98 nm (width), 816 nm (depth). The results show that the uniformity of nano-channels fabricated by thermal nano-imprinting and Parylene deposition is good.

In order to verify the repeatability of the experiment, the process was repeated three times. Figure 8 shows the results of three experiments on the nano-channels after 0.1 g Parylene deposition. The widths of the nano-channels are 48 nm, 36 nm, and 68 nm, respectively, all of which are less than 100 nm, and have a certain repeatability, which is not accidental. The reasons for the non-uniform size of the nano-channels may be due to the size deviation of the SU-8 nano-channels after thermal nano-imprinting. With the increase in Parylene mass, the thickness of the deposited film increased, and the deviation of the original nano-channel size was amplified, resulting in poor uniformity of channel size.

### 3.2. Influence of Deposition Pressure on Deposition Rate and Film Roughness

The roughness of the nano-channels will affect the subsequent packaging and bonding, as well as the function of the nano-fluidic chip. To reduce the roughness of the nano-channel, the surface roughness of the deposited Parylene film should be reduced as much as possible. During the deposition of Parylene film, the deposition rate and roughness of the Parylene film are affected by the deposition pressure, and the roughness of the thin film will affect the quality of the nano-channels. Figure 9 is a three-dimensional diagram of the surface morphology of Parylene film under different deposition pressures, measured by atomic force microscopy. The maximum roughnes (Rmax) of each film is 17.93 nm, 21.61 nm, 26.28 nm, and 37.54 nm, and the root mean square (RMS) roughness is 2.13 nm, 2.39 nm, 2.82 nm, and 2.87 nm under the deposition pressures of 23 mTorr, 33 mTorr, 43 mTorr, and 53 mTorr.

Figure 10 shows the influence of deposition pressure on deposition rate and film roughness during Parylene deposition. It can be seen that the deposition rate rises with the increase in deposition pressure. This is because the molecular concentration of the Parylene monomer in the cavity is proportional to the deposition pressure, and as the deposition pressure increases, the molecular concentration of the Parylene monomer increases; thus, the deposition rate increases. The Rmax and RMS roughness of Parylene film both increase with the increase in deposition pressure, because the lower the deposition pressure, the lower the molecular concentration of Parylene monomer in the deposition cavity. Therefore, the Parylene monomer has sufficient time to polymerize and be deposited on the substrate surface in an orderly manner, which improves the density and uniformity of the film. When the deposition pressure increases, the density and uniformity of the film decreases, and the roughness increases. Therefore, in order to reduce the roughness of the nano-channels, 23 mTorr was selected as the optimal deposition pressure during the experiment.

Due to the good compatibility between Parylene and SU-8 photoresist, the two materials can be hot bonded. In order to improve the bonding strength, Parylene and SU-8 photoresist were treated with oxygen plasma, respectively, before hot bonding. Figure 11 shows the high aspect ratio nano-channels sealed with SU-8 photoresist, of which the sizes are 98 nm in width and 712 nm in depth with the aspect ratio 7 in Figure 11a, and 46 nm in width and 746 nm in depth with the aspect ratio 16 in Figure 11b.

Before the thermal bonding, the SU-8 photoresist cover plate and Parylene nano-channels were treated with oxygen plasma to improve the surface energy, y, which increases the bonding strength. The bonding strength of nano-channels was measured by the vertical stretching method with a tensile machine (FGS-500TW-SL, EHS Instrument, Fuzhou, China). The bonded nano-channels sample was attached to the PMMA fixture of the stretcher with strong adhesive and stretched vertically upward. When the SU-8–Parylene bonding layer cracks, the maximum tensile force of the stretcher is *F*, and the area of the fracture surface is calculated by scanning measurement. Then the bonding strength *T* of nano-channels is defined as *T* = *F*/*S*. The bonding strength of the nano-channels was 0.78 MPa, which proves that the nano-channels have potential application in nano-fluidic chips.

To sum up, the proposed fabrication method for nano-channels by thermal nano-imprinting and Parylene deposition is simple and low cost. However, during the Parylene deposition process, the thickness of Parylene can only be controlled by controlling the mass of Parylene raw material, and the Parylene film deposited on the side walls and bottoms is uncontrolled. In addition, the uniformity and process repeatability of the nano-channels are still insufficient. Therefore, there need to be improvements in terms of accurate film thickness control and uniformity of nano-channels. The controllability of the deposition of Parylene film and the precision of nano-channels should be further improved by optimizing parameters such as the shape of the nano-channels and the deposition temperature and pressure of the Parylene film.

## 4. Conclusions

In this paper, a low-cost fabrication method for nano-channels is presented. The nano-channels were prepared by thermal nano-imprinting and Parylene deposition. The influence of Parylene mass on the thickness of Parylene film was studied. By controlling the mass of Parylene, nano-channels with different sizes were fabricated, including high aspect ratio nano-channels. The effects of deposition pressure on the deposition rate and roughness of Parylene film were studied, and found to increase with the growth of deposition pressure. The optimized deposition pressure of 23 mTorr was obtained to reduce the surface roughness of the nano-channels. The final nano-channels were sealed by SU-8 photoresist by thermal bonding. This fabrication method can realize the preparation of nano-channels under 100 nm with a great aspect ratio, of more than 15, and it is simple, low cost, and does not require expensive equipment.

## Figures and Tables

**Figure 1 micromachines-14-01430-f001:**
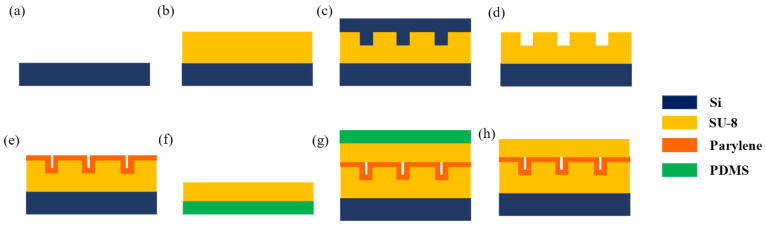
Fabrication process of nano-channels; (**a**) silicon substrate, (**b**) spin-coating SU-8 photoresist layer on silicon, (**c**) thermal nano-imprinting by silicon nano-mold, (**d**) removing silicon nano-mold, (**e**) Parylene deposition, (**f**) spin-coating SU-8 photoresist layer on PDMS, (**g**) thermal bonding, and (**h**) removing PDMS.

**Figure 2 micromachines-14-01430-f002:**
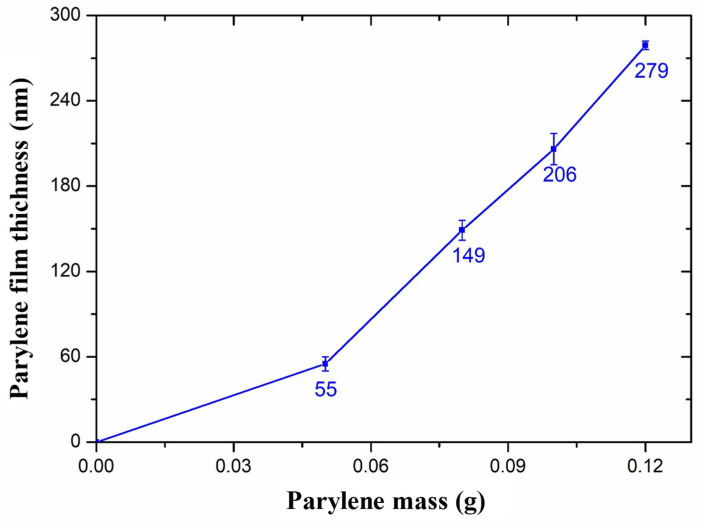
Influence of Parylene mass on film thickness.

**Figure 3 micromachines-14-01430-f003:**
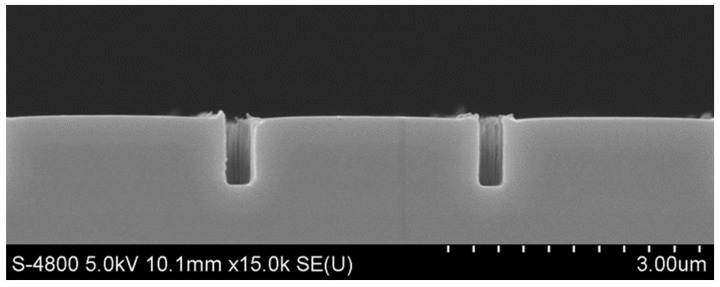
SEM image of SU-8 nano-channels.

**Figure 4 micromachines-14-01430-f004:**
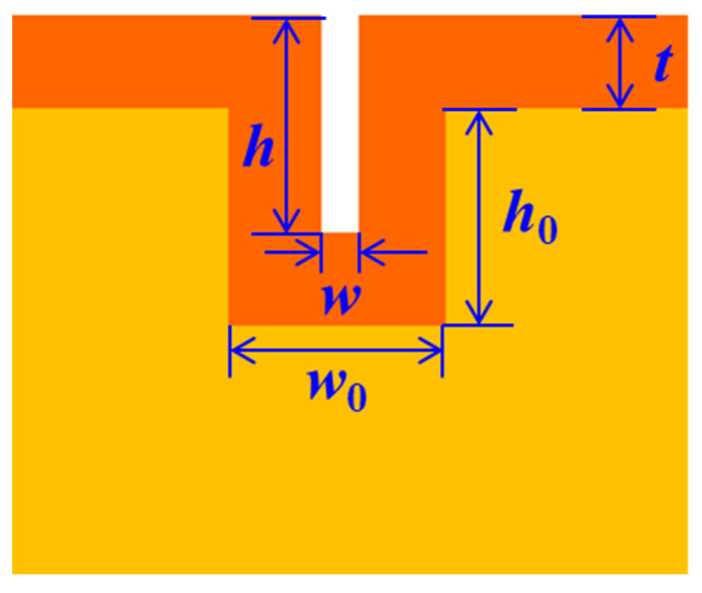
Diagram of nano-channel size.

**Figure 5 micromachines-14-01430-f005:**
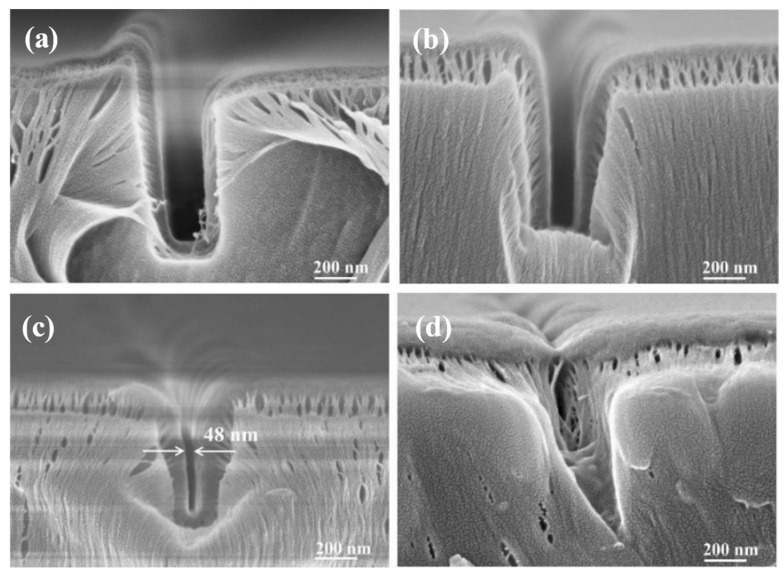
SEM images of nano-channels deposited with different Parylene masses, (**a**) 0.05 g, (**b**) 0.08 g, (**c**) 0.10 g, (**d**) 0.12 g.

**Figure 6 micromachines-14-01430-f006:**
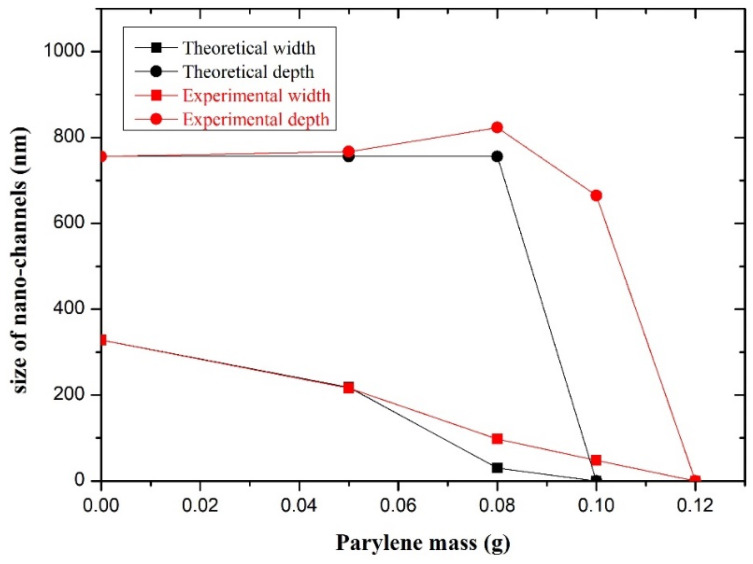
Influence of Parylene mass on size of nano-channels.

**Figure 7 micromachines-14-01430-f007:**
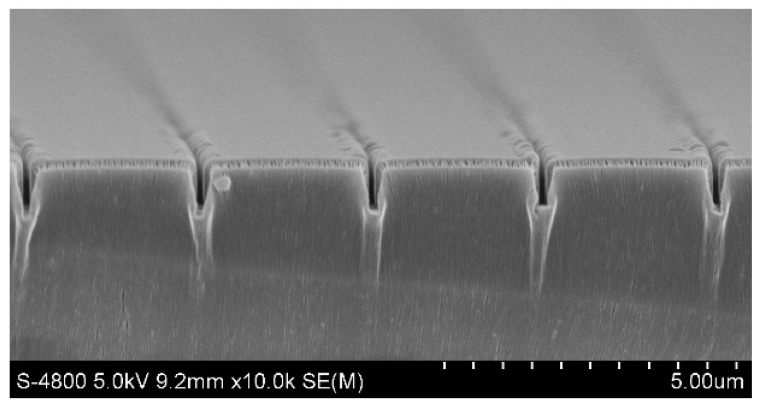
SEM images of nano-channels deposited with 0.08 g Parylene.

**Figure 8 micromachines-14-01430-f008:**
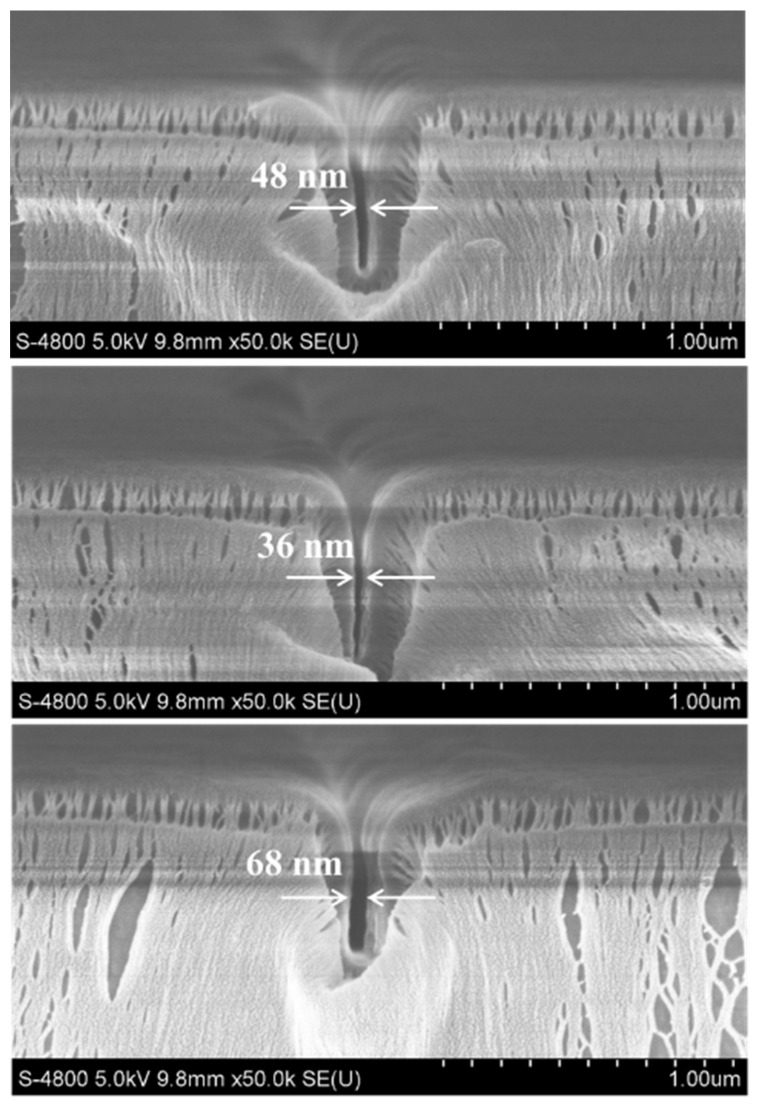
SEM images of nano-channels deposited with 0.1 g Parylene.

**Figure 9 micromachines-14-01430-f009:**
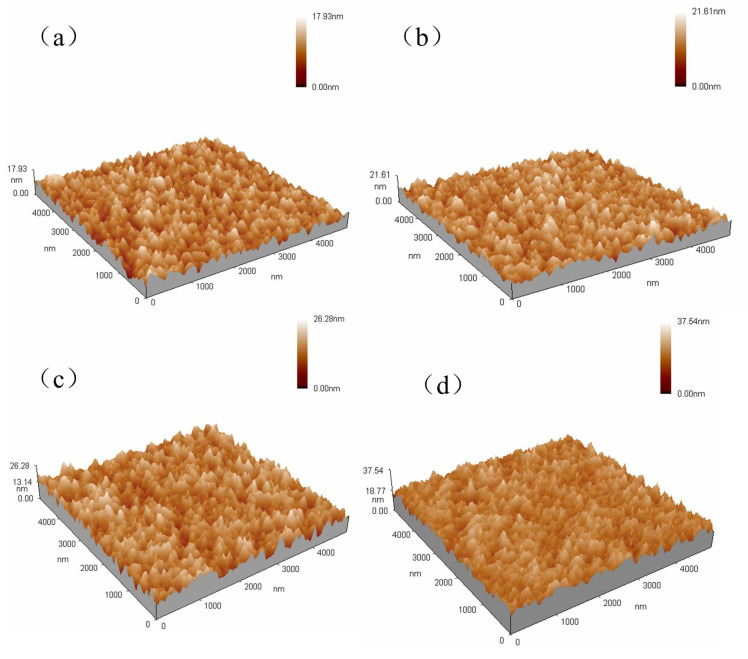
Surface morphology of Parylene film under different deposition pressures, (**a**) 23 mTorr, (**b**) 33 mTorr, (**c**) 43 mTorr, (**d**) 53 mTorr.

**Figure 10 micromachines-14-01430-f010:**
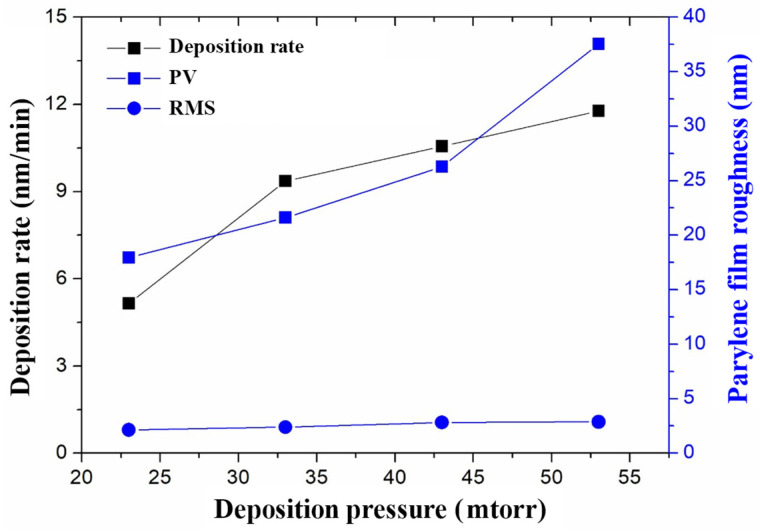
Influence of deposition pressure on deposition rate and film roughness.

**Figure 11 micromachines-14-01430-f011:**
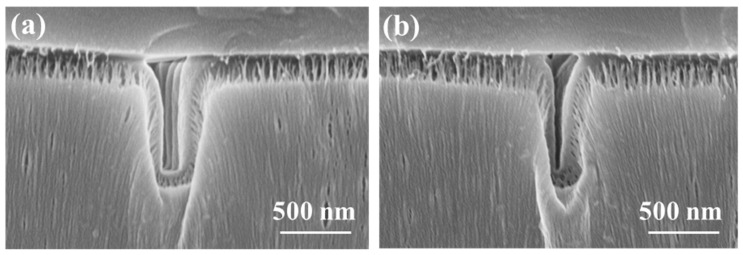
SEM image of nano-channels after thermal bonding, (**a**) 0.08 g Parylene deposition, and (**b**) 0.1 g Parylene deposition.

**Table 1 micromachines-14-01430-t001:** The deposition parameters of Parylene.

Deposition Temperature	Chamber Vacuum	Deposition Pressure
650 °C	8 mTorr	23 mTorr, 33 mTorr, 43 mTorr, 53 mTorr

## Data Availability

The authors confirm that the data supporting the findings of this study are available within the article.

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
