# Peer review of "Fabrication of High Aspect Ratio Nano-Channels by Thermal Nano-Imprinting and Parylene Deposition"

_micromachines, 2023, doi:10.3390/mi14071430_

Round 1

Reviewer 1 Report

You described Fabrication of High Aspect Ratio nano-channels by Thermal Nano-imprinting lithography technology.

It's many interesting paper, however, you should be described some parameters to publish your paper as follows; 

1. It's a little bit short abstract and conclusions for your article. 

2. In abstract, you should more describe how did you reduce the width and increase the height, simultaneously?

 How did you control the thickness in side wall of nano-channels? In fig.5, the deposited thickness of Parylene were very non-uniformly deposited at side wall and bottom areas. 

You suggested 48nm gap of nano-channel, you have to suggest the pattern reproducibility and uniformity on the substrate. 

3. You have to describe more detail information i.e. maker, model etc. of the SU-8 205 and Parylene materials.

4. In Fig.5(a) and (b), there are not equal to the thickness of Parylene in left and right side wall. 

   It seems to be some problems in Parylene deposition.  In general, parylene is able to make good coating uniformity on the substrate. You have to describe these problems.  

Author Response

Reviewer 1

Comments and Suggestions for Authors

You described Fabrication of High Aspect Ratio nano-channels by Thermal Nano-imprinting lithography technology.

It's many interesting paper, however, you should be described some parameters to publish your paper as follows;

  1. It's a little bit short abstract and conclusions for your article.

Response: Thank you for your comments. We have made some additions in abstract and conclusions, as follows:

Abstract

A low-cost fabrication of high aspect ratio nano-channels by thermal nano-imprinting and Parylene deposition is proposed. The SU-8 photoresist nano-channels were first manufactured by thermal nano-imprinting and Parylene deposition was carried out to reduce the width of the nano-channels and increase the aspect ratio. During the process, the side wall of SU-8 nano-channels was covered with the Parylene film, reducing the width of the nano-channel, and the depth of the channel increases due to the thickness of the Parylene film deposited on the surface of SU-8 nano-channels more than that at the bottom. The influence of Parylene mass on size of nano-channels was studied by theoretical analysis and experiments, and the deposition pressure of Parylene was optimized. The final high aspect ratio nano-channels are 46 nm in width and 746 nm in depth, of which the aspect ratio is 16. This simple and efficient method paves the way for high aspect ratio nano-channels.

Conclusion

In this paper, a low-cost fabrication method for nano-channels is presented. The nano-channels were prepared by thermal nano-imprinting and Parylene deposition. The influence of parylene mass on the thickness of Parylene film was studied. By controlling the mass of Parylene, nano-channels with different sizes were fabricated, including high aspect ratio nano-channels. The effect of deposition pressure on the deposition rate and roughness of Parylene film was studied, which increase with the growth of deposition pressure. The optimized deposition pressure of 23 mTorr was obtained to reduce the surface roughness of the nano-channels. The final nano-channels were sealed by SU-8 photoresist by thermal bonding. This fabrication method can realize the nano-channel preparation with a great aspect ratio more than 15, which is simple, low-cost, and expensive equipment independence.

  1. In abstract, you should more describe how did you reduce the width and increase the height, simultaneously?

Response: Thank you for your comments. We added specific description in the abstract and highlighted in yellow, as follows:

During the process, the side wall of SU-8 nano-channels was covered with the Parylene film, reducing the width of the nano-channel, and the depth of the channel increases due to the thickness of the Parylene film deposited on the surface of SU-8 nano-channels more than that at the bottom.

How did you control the thickness in side wall of nano-channels? In fig.5, the deposited thickness of Parylene were very non-uniformly deposited at side wall and bottom areas.

Response: Thank you for your comments.

The cross-section of the parylene nano-channels in figure 5 is the sample preparation achieved by manual breaking. Due to the high fracture elongation of Parylene, the Parylene film breaks and extends during the preparation, and parts of the nano-channels were also covered by it. Therefore, it is impossible to accurately determine the thickness of the film deposited on the side wall and bottom through figure 5.

In the deposition process, we control the thickness of the Parylene film by controlling the mass of Parylene raw material, as shown in figure 2. Theoretically, if the thickness of Paylene deposited on the side wall and bottom is the same as the thickness of the surface, the final size of the nano-channels should be the theoretical curve in figure 6. But the actual experimental curve is different from the theoretical curve, mainly because the thickness of the film deposited on the side wall and bottom of Payrlene is smaller than that deposited on the surface. Moreover, the thickness of the film deposited on the side wall is usually smaller than that deposited on the bottom, resulting in the difference between theory and practice. Here we do not have a way to accurately control the thickness of the Parylene film, but by controlling the mass of Parylene, different sizes of nano-channels can be made, including the high aspect ratio of nano-channels.

You suggested 48nm gap of nano-channel, you have to suggest the pattern reproducibility and uniformity on the substrate.

Response: Thank you for your comments.

We conducted three experiments, and the specific experimental results are shown in the figure below. The widths of the nano-channel are 48 nm, 36 nm and 68 nm respectively, all of which are less than 100 nm, and have certain repeatability, which is not accidental. However, the controllability of the nano-channel size really needs to be improved.

Due to the large fracture elongation of Parylene, the fracture elongation of Parylene film occurs during the preparation of cross-section electron microscopy samples. The thicker the film, the more serious the fracture elongation phenomenon will be, and even most of nano-channels are blocked, which causes certain difficulties in our study of the uniformity of nano-channles. The following figure shows the SEM interface morphology of multiple nano-channels after growing 0.08 g Parylene, and it can be seen that the uniformity of the nano-channels is good.

  1. You have to describe more detail information i.e. maker, model etc. of the SU-8 205 and Parylene materials.

Response: Thank you for your comments.

In the experimental steps, we add the manufacturers and models of the two materials as follows: SU-8 photoresist (MicroChem, SU-8 2015)   Parylene N (Suzhou Yacoo Science Co., Ltd)

The above mentioned were added in the revised manuscript and highlighted in yellow.

  1. In Fig.5(a) and (b), there are not equal to the thickness of Parylene in left and right side wall.

It seems to be some problems in Parylene deposition.  In general, parylene is able to make good coating uniformity on the substrate. You have to describe these problems. 

Response: Thank you for your comments.

Since the left and right sides of the SU-8 nano-channels fabricated by thermal nano-imprinting in figure 3 are not completely symmetrical, the thickness of the left and right Parylene films after growing Parylene is different. The method mentioned in this paper is low-cost and simple, but it still needs to be improved in terms of accurate film thickness control and uniformity. In the future, we will further improve the controllability of the deposition of Parylene film and the precision of the nano-channels by optimizing parameters such as the shape of the nano-channels, the growth temperature or pressure of the Parylene film.

The above mentioned were added in the revised manuscript and highlighted in yellow.

Reviewer 2 Report

This paper presented a method to fabricate high aspect ratio nano-channels. The authors’ fabrication method will contribute to basic technology for nano-fabrication. The fabrication method, which is easy, rapid, and low-cost, is useful for micro-fluidics and nano-fluidics researchers. Therefore, I recommend the publication of the manuscript after a minor revision.

Suggestions:

1) The experimental parameters should be more detailed and clear, such as the parameters of SU-8 photoresist casting, and the parameters of CVD growing parylene film, etc.

2) In the SEM images shown in Figure 5, Why does the film fall off and deform?

3) In the paper, SU-8 photoresist was used to bond Parylene. How to achieve complete bonding and ensure sufficient bonding strength.

Author Response

Reviewer 2

Comments and Suggestions for Authors

This paper presented a method to fabricate high aspect ratio nano-channels. The authors’ fabrication method will contribute to basic technology for nano-fabrication. The fabrication method, which is easy, rapid, and low-cost, is useful for micro-fluidics and nano-fluidics researchers. Therefore, I recommend the publication of the manuscript after a minor revision.

Suggestions:

1) The experimental parameters should be more detailed and clear, such as the parameters of SU-8 photoresist casting, and the parameters of CVD growing parylene film, etc.

Response: Thank you for your comments. We added specific experimental parameters in the revised manuscript and highlighted in yellow, as follows:

First, the SU-8 photoresist (MicroChem, SU-8 2015) was spin-coated on the silicon substrate at a speed of 4000 rpm for 30 s, followed by pre-baking at 85 oC for 30 min (Fig. 1(a) and (b)). Then the SU-8 photoresist was thermal imprinted by a silicon nano-mold in an oven at 85 oC for 5 min (Fig. 1(c)), and the silicon nano-mold was removed after cooling (Fig. 1(d)). After that, the high aspect ratio nano-channels were produced by chemical vapor deposited (CVD) with Parylene N (Suzhou Yacoo Science Co., Ltd) on the SU-8 nano-channels, the specific deposition parameters are shown in Table 1. The next step, SU-8 photoresist 2015 was spin-coated on the PDMS substrate, followed by pre-baking at 85 oC for 30 min to get a cover SU-8 layer (Fig. 1(f)). And the nano-channels were thermal bonding with the cover SU-8 layer at 50 °C for 5 min. (Fig. 1(g)). Finally, the PDMS was removed from the cover SU-layer, and the high aspect ratio nano-channels were fabricated (Fig. 1(h)).

Table 1. The deposition parameters of parylene

Deposition temperature

Chamber vacuum

Deposition 

pressure

650 oC

8 mTorr

23 mTorr、33 mTorr、43 mTorr、53 mTorr

2) In the SEM images shown in Figure 5, Why does the film fall off and deform?

Response: Thank you for your comments.

Due to the high fracture elongation of Parylene, the fracture elongation of Parylene thin film as shown in the figure occurs during the preparation of cross-section electron microscope samples.

3) In the paper, SU-8 photoresist was used to bond Parylene. How to achieve complete bonding and ensure sufficient bonding strength.

Response: Thank you for your comments.

Before the thermal bonding, the SU-8 photoresist cover plate and Parylene nano-channels were treated with oxygen plasma to improve the surface energy, which will increase the bonding strength and ensure the complete bonding.

The above mentioned were added in the revised manuscript and highlighted in yellow.

Round 2

Reviewer 1 Report

You described Fabrication of High Aspect Ratio nano-channels by Thermal Nano-imprinting lithography technology.

It's many interesting paper, however, you should be described some parameters to publish your paper as follows; 

1. How did you control the thickness in side wall of nano-channels? In fig.5, the deposited thickness of Parylene were very non-uniformly deposited at side wall and bottom areas.

2. In Fig.5(a) and (b), there are not equal to the thickness of Parylene in left and right side walls.  It seems to be some problems in Parylene deposition.  In general, Parylene is able to make good coating uniformity on the substrate. Does these results are equally generated in all of your experiments?   

3. You suggested 48nm gap (46nm in abstract) of nano-channel, you have to describe the  pattern reproducibility and uniformity on the substrate. If it is difficult to describe pattern reproducibility and uniformity, it would be fine if you could explain the implementation probability of these nano-gaps.

Author Response

Reviewer 1

You described Fabrication of High Aspect Ratio nano-channels by Thermal Nano-imprinting lithography technology.

It's many interesting paper, however, you should be described some parameters to publish your paper as follows;

  1. How did you control the thickness in side wall of nano-channels? In fig.5, the deposited thickness of Parylene were very non-uniformly deposited at side wall and bottom areas.

Response: Thank you for your comments.

The cross-section of the parylene nano-channels in figure 5 is the sample preparation achieved by manual breaking. Due to the high fracture elongation of Parylene, the Parylene film breaks and extends during the preparation, and parts of the nano-channels were also covered by it. Therefore, the thickness of the Parylene film deposited on the side wall and bottom as observed in figure 5 is not accurate.

In the deposition process, we control the thickness of the Parylene film by controlling the mass of Parylene raw material, as shown in figure 2. Theoretically, if the thickness of Paylene deposited on the side wall and bottom is the same as the thickness of the surface, the final size of the nano-channels should be the theoretical curve in figure 6. But the actual experimental curve is different from the theoretical curve, mainly because the thickness of the film deposited on the side wall and bottom of Payrlene is smaller than that deposited on the surface. Moreover, the thickness of the film deposited on the side wall is usually smaller than that deposited on the bottom, resulting in the difference between theory and practice. Here we do not have a way to accurately control the thickness of the Parylene film, but by controlling the mass of Parylene, different sizes of nano-channels can be made, including the high aspect ratio of nano-channels.

The above mentioned were added in the revised manuscript and highlighted in yellow.

  1. In Fig.5(a) and (b), there are not equal to the thickness of Parylene in left and right side walls. It seems to be some problems in Parylene deposition. In general, Parylene is able to make good coating uniformity on the substrate. Does these results are equally generated in all of your experiments?  

Response: Thank you for your comments.

On the one hand, due to the high fracture elongation of Parylene, the Parylene film breaks and extends during the preparation, and parts of the nano-channels were also covered by it. Therefore, the thickness of the Parylene film deposited on the side wall and bottom observed in the SEM images is not accuracy. Thus, it is not completely concluded that the thickness of the left and right sides is not consistent.

On the other hand, maybe because the left and right sides of the SU-8 nano-channels fabricated by thermal nano-imprinting in figure 3 are not completely symmetrical, the thickness of the left and right Parylene films after growing Parylene is different.

In addition, although Parylene is able to make good coating uniformity on the substrate. The probability that Parylene molecules enter the nano-channels and deposited on the side wall and bottom may be less than that on the surface and affected with the change of the size of the nano-channel, resulting in uneven thickness.

To sum up, the method mentioned in this paper is low-cost and simple, and can fabricate nano-channels under 100 nm and with high aspect ratio, but it still needs to be improved in terms of accurate film thickness control and uniformity. In the future, we will further improve the controllability of the deposition of Parylene film and the precision of the nano-channels by optimizing parameters such as the shape of the nano-channels, the growth temperature or pressure of the Parylene film.

The above mentioned were added in the revised manuscript and highlighted in yellow.

  1. You suggested 48nm gap (46nm in abstract) of nano-channel, you have to describe the pattern reproducibility and uniformity on the substrate. If it is difficult to describe pattern reproducibility and uniformity, it would be fine if you could explain the implementation probability of these nano-gaps.

Response: Thank you for your comments.

We conducted three experiments, and the specific experimental results are shown in the figure below. The widths of the nano-channel are 48 nm, 36 nm and 68 nm respectively, all of which are less than 100 nm, and have certain repeatability, which is not accidental. The reasons for the non-uniform size of nano-channels maybe due to the size deviation of SU-8 nano-channels after thermal nano-imprinting. With the increase of Parylene mass, the thickness of the deposited film increases, the deviation of the original nano-channel size is amplified, resulting in poor uniformity of channel size.

Due to the large fracture elongation of Parylene, the fracture elongation of Parylene film occurs during the preparation of cross-section electron microscopy samples. The thicker the film, the more serious the fracture elongation phenomenon will be, and even most of nano-channels are blocked, which causes certain difficulties in our study of the uniformity of nano-channles. The following figure shows the SEM interface morphology of multiple nano-channels after growing 0.08 g Parylene, and it can be seen that the uniformity of the nano-channels is good.

The above mentioned were added in the revised manuscript and highlighted in yellow.

Round 3

Reviewer 1 Report

It's many interesting paper, however, you should be described some parameters to publish your paper as follows; 

1. You suggested 48nm gap (46nm in abstract) of nano-channel. You have to modify as 48nm according to Fig.8.